# Vaginal Dinoprostone Insert versus Cervical Ripening Balloon for Term Induction of Labor in Obese Nulliparas—A Randomized Controlled Trial

**DOI:** 10.3390/jcm11082138

**Published:** 2022-04-11

**Authors:** Roy Lauterbach, Dikla Ben Zvi, Haneen Dabaja, Ragda Zidan, Naphtali Justman, Dana Vitner, Ron Beloosesky, Nadir Ghanem, Yuval Ginsberg, Yaniv Zipori, Zeev Weiner, Nizar Khatib

**Affiliations:** 1Department of Obstetrics and Gynecology, Rambam Health Care Campus, Haifa 3109601, Israel; d_benzvi@rambam.health.gov.il (D.B.Z.); ha_dabaja@rambam.health.gov.il (H.D.); ragdazid@gmail.com (R.Z.); n_justman@rambam.health.gov.il (N.J.); d_vitner@rambam.health.gov.il (D.V.); ronbel3@gmail.com (R.B.); n_ganem@rambam.health.gov.il (N.G.); yuvalginsberg@gmail.com (Y.G.); y_zipori@rambam.health.gov.il (Y.Z.); z_weiner@rambam.health.gov.il (Z.W.); khatibnizar@yahoo.com (N.K.); 2Bruce Rappaport Faculty of Medicine, Technion-Israel Institute of Technology, Haifa 3200003, Israel

**Keywords:** anxiety, body mass index, cervical ripening balloon, induction, obesity, satisfaction

## Abstract

Data regarding the preferred induction method in women with obesity is scarce. The current study was aimed at comparing pharmacological and mechanical induction in this population. This prospective randomized controlled trial was conducted between 2016–2020, in nulliparas with a pre-pregnancy body mass index >30. Inclusion criteria were singleton-term pregnancies, bishop score < 5, and indication for induction. Patients were randomized to induction by a cervical ripening balloon (CRB) or a 10 mg vaginal dinoprostone insert. The primary outcome was delivery rate within 24 h. Secondary outcomes included time to delivery, cesarean section rate, maternal and neonatal outcomes, satisfaction, and anxiety. The study population comprised of 83 women in the CRB group and 81 in the dinoprostone group. There was a significant difference in delivery rates within 24 h and time to delivery between the dinoprostone and CRB groups (45% vs. 71%, *p* = 0.017 and 49.3 ± 6.8 h vs. 23.5 ± 5.9 h, *p* = 0.003, respectively). There were no differences in cesarean delivery rates or maternal and neonatal outcomes, though CRB induction was associated with a significantly lower rate of tachysystole. Induction with CRB was accompanied by higher satisfaction and lower anxiety. In summary, CRB induction is associated with shorter time to delivery, higher satisfaction, and lower anxiety compared to PGE2 in women with obesity, without compromising maternal or neonatal outcomes.

## 1. Introduction

The prevalence of obesity among reproductive-age women is increasing worldwide. In pregnant women, it is estimated to be as high as 30%. Women with obesity have an approximately 2.5-fold higher risk of pregnancy complications. The severity of obesity during pregnancy is correlated with increased risk for both maternal and perinatal complications [1,2].

Higher maternal pre-pregnancy body mass index (BMI) and gestational weight gain are associated with higher risks of antepartum complications, including early pregnancy loss, gestational hypertensive disorders, gestational diabetes, prolonged pregnancies, macrosomia, and preterm birth [2,3]. Furthermore, intrapartum complications may include difficulties with regional anesthesia, prolonged duration of the 1st stage of labor, shoulder dystocia, and cesarean section (CS) [4,5,6]. Postpartum related complications may include venous thromboembolism, depression, trouble losing weight, and future cardiovascular complications [7,8,9].

Due to their tendency for prolonged pregnancies, women with obesity are prone to induction of labor (IOL). Unfortunately, obesity is also associated with a higher rate of failed IOL, requiring either additional IOL with prostaglandins or a cervical ripening balloon (CRB) and augmentation or a CS, specifically in nulliparous women [10]. Studies regarding optimal induction methods in women with obesity have demonstrated conflicting results and have been mostly retrospective in nature. A recent retrospective study of 192 nulliparous overweight and obese women found that neither pharmacological IOL with dinoprostone or misoprostol, nor mechanical IOL with a CRB significantly impacted the induction to birth time or the CS rate [11]. In another retrospective study, misoprostol led to a higher rate of successful IOL and a lower CS rate compared to dinoprostone [12]. Women with obesity that received misoprostol took longer to deliver by up to 4 h and had a higher rate of CS for all indications compared to non-obese women [13]. This may be explained by a mechanism of poorer myometrial contractility, as previously demonstrated [14].

Due to the nature of the growing obesity epidemic and the need to clarify important management points in the obese population, the aim of the current study was to determine the preferred IOL method for nulliparous women with pre-pregnancy obesity (BMI > 30).

## 2. Materials and Methods

### 2.1. Study Design

This was a single center, open label, randomized controlled trial designed to compare mechanical IOL to pharmacological IOL in women with pre-pregnancy obesity. The study received institutional review board approval (0192-19-RMB) and was registered at clinicaltrial.gov (NCT03033264). The study was conducted between June 2016 and July 2020. All participants signed informed consent after receiving a thorough explanation regarding the study.

### 2.2. Study Population

Inclusion criteria for the study were nulliparous women with a pre-pregnancy and pre-induction BMI > 30, over 18 years and under 44 years, singleton pregnancy, vertex presentation, term ≥39 + 0 and ≤41 + 0 weeks’ gestation (based on early ultrasound performed at up to 10 weeks’ gestation), and candidates for IOL due to either post-date, diabetes in pregnancy, hypertension related complications of pregnancy (preeclampsia and pregnancy induced hypertension), decreased fetal movements after 39 weeks’ gestation, or maternal request after 39 weeks’ gestation. Exclusion criteria were Bishop score ≥ 6, non-vertex presentation, premature rupture of membranes, non-reassuring fetal tracings, intrauterine growth restriction, intrauterine fetal demise, previous uterine scar, placenta previa, suspected/confirmed genital herpes infection, known HIV seropositivity, and suspected congenital abnormalities.

### 2.3. Study Intervention

Prior to the decision regarding IOL method, digital examination to determine the Bishop score and fetal cardiotocography to verify FHR normality were performed. Participants were randomly allocated in a 1:1 ratio with a block size of six to one of the two treatment groups by a secure, computer-generated, online centralized web-based system. The first treatment group was induced with a silicone double lumen cervical ripening balloon (CRB) (Cook Cervical Ripening Balloon, Cook Medical Europe, Limerick, Ireland). The second treatment group was induced with a slow-release vaginal insert containing 10 mg of Dinoprostone (prostaglandins PGE2 Propess, Ferring SAS, Gentilly, France). Regarding storage directives, the silicone double lumen balloon was stored in a dry place, away from light, and the slow-release vaginal insert was stored in a freezer at a temperature between −20 °C and −10 °C. Insertion of either treatment was performed by trained physicians accustomed to using both options in their daily practice. Removal of the CRB was performed 12 h after insertion if self-expulsion did not occur, while removal of the dinoprostone vaginal insert was performed 24 h after insertion if self-expulsion did not occur. In the CRB group, patients underwent fetal heart rate monitoring immediately after balloon insertion and subsequently every 6–8 h, while in the dinoprostone group, tracing was performed up to 2 h after insert placement and every 6 h afterwards. After the removal or self-expulsion of the IOL agents, patients underwent a vaginal examination and were transferred to the delivery ward if their Bishop score was over 5, where they either continued labor spontaneously or continued augmentation, if necessary, by oxytocin infusion and/or amniotomy.

### 2.4. Study Outcomes

The primary outcome was delivery rate within 24 h. Secondary outcomes included CS rate, induction-to-delivery interval, total time to delivery from induction commencement, need for augmentation with oxytocin, uterine tachysystole, clinical chorioamnionitis and suspected maternal postpartum infection based on presence of a fever > 38 °C with or without malodourous vaginal discharge, uterine sensitivity upon palpation or fetal tachycardia, surgical site infection, postpartum hemorrhage, blood transfusion, obstetric anal sphincter injury, uterine revision, shoulder dystocia, length of hospitalization, satisfaction and anxiety levels, Apgar < 5 at 5 min, arterial cord pH < 7.1, admission to the NICU, need for oxygen supplementation, hypoglycemia, sepsis, and Erb’s palsy.

Patient satisfaction was evaluated using the satisfaction with childbirth experience Six Simple Questions (SSQ). The SSQ is scored on a 7-point scale. The score ranges from 7–42, with higher scores indicating higher patient satisfaction [15].

Patient Anxiety was evaluated using the State-Trait Anxiety Inventory (STAI) questionnaire. The STAI is comprised of 20 questions with a 4-point scale. The score ranges from 20–80, with higher scores indicating lower patient anxiety [16].

### 2.5. Statistical Analysis

The sample size was calculated based on our hypothesis that mechanical IOL will lead to a 20% advantage in delivery rates within 24 h compared with pharmacological IOL. Our hypothesis was based on retrospective data collected regarding women with obesity that delivered at our center after IOL between the years 2010 and 2015. To detect a 20% difference in the primary outcome with a power of 80% and a two-tailed type I error of 5%, we required the inclusion of a total of 154 women (77 in each group).

Statistical analysis was performed according to the intention-to-treat principle. Continuous variables were calculated as mean ± SD and compared using Student’s *t*-test or the non-parametric Mann–Whitney test as appropriate. Categorical variables were calculated as rate (percentage) and compared with chi-squared or Fisher’s exact test as appropriate. The 27.0 SPSS version for Windows (SPSS, Chicago, IL, USA) was used for statistical analysis and data management.

## 3. Results

Of the 348 women screened, 214 were found to be eligible for the study, of which 164 agreed to participate and were randomized to either PGE2 or CRB IOL and included in the final analysis (81 in the PGE2 group and 83 in the CRB group) (Figure 1). The data were analyzed on an intention-to-treat basis. Three of the patients randomized to the PGE2 group underwent sequential CRB induction after PGE2 failure at their request. Four of the patients randomized to the CRB group underwent sequential PGE2 induction attempt after CRB failure at their request. All seven of these patients were included in the final data analysis since there was no deviation from the randomized intended primary IOL method. Baseline demographic, medical and obstetric characteristics were similar in both groups (Table 1). The rate of delivery within 24 h of induction was significantly higher in the CRB group compared to the PGE2 group (71.1% vs. 45.6%, odds ratio (OR) (95% confidence interval (CI)) 1.56 (1.23–2.89), *p* = 0.017). Furthermore, patients in the CRB group delivered significantly earlier, by more than a day compared to patients in the PGE2 group (23.5 ± 5.9 vs. 49.3 ± 6.8 h, OR (95% CI) 2.09 (1.44–3.18), *p* = 0.003) (Table 2).

The vaginal delivery rate and rate of additional augmentation wasn’t significantly different between the CRB and PGE2 groups (81.9% vs. 80.2%, OR (95% CI) 1.02 (0.69–1.44), *p* = 0.58 and 85.5% vs. 79%, OR (95% CI) 1.36 (0.91–1.72), *p* = 0.31, respectively). There was, however, a lower rate of CS performed due to non-reassuring fetal heart rate tracing in the CRB group compared to the PGE2 group (26.7% vs. 50%, OR (95% CI), 0.53 (0.37–0.83), *p* = 0.039). Similarly, the incidence of tachysystole was significantly lower in the CRB group compared to the PGE2 group (4.8% vs. 23.4%, OR (95% CI), 0.21 (0.17–0.65), *p* < 0.0001). 

There were no significant differences in additional maternal outcomes—including rate of postpartum hemorrhage, chorioamnionitis, surgical site infection, revisio uteri, and shoulder dystocia—between the CRB and PGE2 groups (Table 2).

There were no significant differences in neonatal outcomes—including rates of Apgar < 5 at 5 min, arterial cord pH < 7.1, admission to the NICU, need for oxygen supplementation, hypoglycemia, sepsis, and Erb’s palsy—between the CRB and PGE2 groups (Table 3).

Patient satisfaction was significantly higher in the CRB group compared to the PGE2 group (36.41 ± 4.02 vs. 27.59 ± 3.38, OR (95% CI), 1.32 (1.04–1.67), *p* = 0.025). Furthermore, patient anxiety was significantly lower in the CRB group compared to the PGE2 group (71.03 ± 6.78 vs. 58.42 ± 4.56, OR (95% CI), 1.21 (1.02–1.44), *p* = 0.043).

## 4. Discussion

### Principal Findings

The current RCT compared pharmacological IOL with PGE2 and mechanical IOL with a double-lumen CRB for singleton term pregnancies in nulliparous women with a pre-pregnancy BMI > 30. IOL with a CRB led to higher delivery rates within 24 h, reduced time to delivery, and reduced need for additional augmentation. Rate of vaginal delivery remained unchanged. Patient satisfaction was higher with lower anxiety levels in the CRB group.

Previous studies regarding the efficacy and safety of PGE2 vs. CRB term IOL have shown no significant differences in either maternal or neonatal outcomes [17]. Regarding IOL in patients with obesity, the majority of studies that addressed this specific issue were retrospective in nature and included cohorts of between 100–200 patients [18,19,20,21,22,23,24,25]. While these studies found no difference in induction to delivery intervals with regard to IOL agents used, patient satisfaction was mentioned in a single study that compared mechanical IOL in obese compared to non-obese women, where no difference was found in patient satisfaction between the two groups [17]. The mainstay of IOL superiority among patients with obesity and overweight patients is that there are no differences in obstetric, maternal, or neonatal outcomes in women induced by PGE2, PGE1, or CRB [18,19,20,21,22,23,24,25,26,27].

Obesity is a known risk factor for prolonged pregnancy, with associated higher rates of IOL, IOL failure, and CS. Obesity is also associated with longer labor times and higher costs for delivery due to the additional complications and extra-maintenance [28,29,30]. With the increasing prevalence of obesity among reproductive-age women, there is a genuine need to optimize and personalize the IOL process in this population. The current study demonstrated not only the importance of the correlation between the IOL method and time to delivery, maternal, and neonatal outcomes, but also demonstrated the effect of determining optimal methods of IOL on patient satisfaction and anxiety. These effects may influence patients’ decision making regarding the site of their future deliveries, in addition to various psychological effects and factors that may have significant financial and emotional impact on healthcare providers. Thus, in a population of obese, otherwise low-risk, nulliparous women, IOL with CRB is considered a safe and beneficial option for both patient and healthcare providers.

Obese multigravidas seem to have the same risk for IOL as patients with a normal BMI [19]. For this reason and the previously mentioned reasons, the current study concentrated on nulliparous obese women. Since the current study showed the superiority of IOL with CRB over PGE2 and due to previous retrospective data suggesting some superiority of misoprostol over PGE2 for IOL in patients with obesity [12], it would be beneficial to investigate the additional agents for IOL in women with obesity at term in the future. Furthermore, while the current study singled out the population of women with obesity and no additional high-risk features, this population has a high rate of pregnancy comorbidities; further research regarding the optimal method for IOL in the entire obese pregnant population—including possible stratification by obesity classes—is warranted.

The increased satisfaction and decreased anxiety in the CRB group may be associated with the shorter time to delivery and the shorter interval of uncertainty patients experienced during their labor experience Figure 2. Furthermore, this may be a result of the longer expected intervals the different primary inductions may take, i.e., 24 h for the PGE2 insert vs. 12 h for the CRB. Due to the fact that patient blinding was impossible, this may have impacted results of patient satisfaction and anxiety; as such, this should be mentioned as a possible limitation of these outcomes’ validity.

The strengths of the current study include the randomized controlled trial design, the homogeneity of the study population (low-risk women with obesity without additional pregnancy comorbidities), and the emphasis on patient satisfaction and anxiety as part of the attempt to evaluate the whole experience of IOL and labor (not just the standard maternal and neonatal outcomes). 

Study limitations include the lack of blinding, which was not feasible in a study of this design. In addition, as insertion of a CRB is done under supervision and may still pose a challenge to the physician, the insertion and correct placement of PGE2 in obese women may be physically challenging as well. These technical difficulties may have some effect on the study results. The generalizability of the results of the current study is limited due to the exclusion of obese women with additional high-risk features. The current study made use of a double-lumen CRB, while in many medical centers, a Foley catheter is the mechanical IOL agent of choice. 

## 5. Conclusions

Our results suggest that IOL with a CRB compared to PGE2 in nulliparous women with obesity was associated with reduced time to delivery, higher patient satisfaction, and lower levels of anxiety, without affecting additional maternal or neonatal adverse outcomes. Further research is warranted to establish the preferred IOL method for nulliparous obese women.

## Figures and Tables

**Figure 1 jcm-11-02138-f001:**
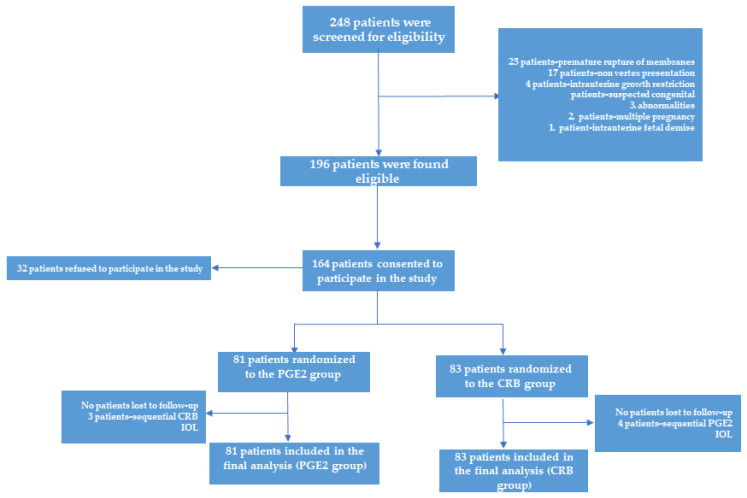
Patient CONSORT flow.

**Figure 2 jcm-11-02138-f002:**
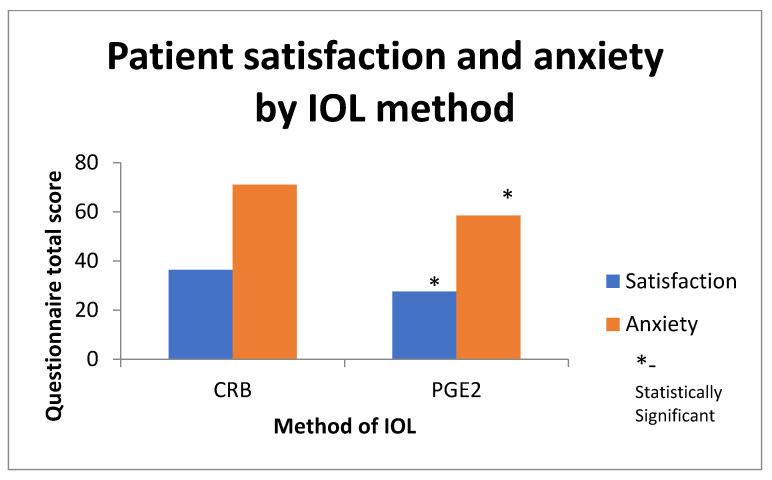
Patient satisfaction and anxiety based on IOL method.

**Table 1 jcm-11-02138-t001:** Baseline demographic, medical, and obstetric information.

	PGE2 (n = 81)	CRB (n = 83)
Age, years, mean ± SD	27.9 ± 4.5	27.3 ± 5.8
GA at induction, weeks, mean ± SD	39.1 ± 1.5	39.3 ± 1.4
Pre-pregnancy BMI, kg/m^2^, mean ± SD	33.5 ± 2.6	34.2 ± 3.1
BMI prior to induction, kg/m^2^, mean ± SD	36.1 ± 4.2	36.4 ± 3.8
Bishop score at admission, median (range)	1 (0–5)	1 (0–5)
Indication for induction, n (%) post date	28 (34.6)	31 (37.3)
Diabetes in pregnancy	19 (23.4)	17 (20.5)
Hypertension in pregnancy	15 (18.5)	19 (22.9)
Decreased fetal movements	14 (17.3)	12 (14.4)
Maternal request	5 (6.2)	4 (4.9)
Rate of diabetes in pregnancy, n (%)	23 (28.4)	21 (25.3)
Rate of hypertension related complications of pregnancy, n (%)	20 (24.7)	22 (26.5)

Abbreviations: PGE—prostaglandin E; CRB—cervical ripening balloon; n—number; SD—standard deviation; GA—gestational age; BMI—body mass index; kg—kilogram; m—meter.

**Table 2 jcm-11-02138-t002:** Maternal outcomes.

	PGE2 (n = 81)	CRB (n = 83)	OR (95% CI)	*p* Value
Delivery within 24 h of induction, n (%)	37 (45.6%)	59 (71.1%)	1.56 (1.23–2.89)	0.017
Time to delivery, hours, mean ± SD	49.3 ± 6.8	23.5 ± 5.9	2.09 (1.44–3.18)	0.003
Vaginal delivery rate, n (%)	65 (80.2%)	68 (81.9%)	1.02 (0.69–1.44)	0.58
CS rate by indication, n (%)	16 (19.8%)	15 (18.1%)	1.01 (0.72–1.48)	0.59
Obstructed Labor	6 (37.5%)	8 (53.3%)	1.42 (0.74–1.78)	0.15
Non-reassuring fetal heart rate	8 (50%)	4 (26.7%)	0.53 (0.37–0.83)	0.039
Maternal request	2 (12.5%)	3 (20%)	1.61 (0.96–1.84)	0.09
Tachysystole, n (%)	19 (23.4%)	4 (4.8%)	0.21 (0.17–0.65)	<0.0001
Need for additional induction, n (%)	4 (4.9%)	3 (3.6%)	1.36 (0.91–1.72)	0.26
Need for augmentation with Pitocin, n (%)	64 (79%)	71 (85.5%)	1.08 (0.81–1.35)	0.31
Postpartum Hemorrhage, n (%)	12 (14.8%)	14 (16.8%)	1.13 (0.75–1.41)	0.63
Chorioamnionitis, n (%)	9 (11.1%)	11 (13.2%)	1.19 (0.68–1.37)	0.67
Shoulder Dystocia, n (%)	5 (6.1%)	6 (7.2%)	1.18 (0.73–1.45)	0.39
Revisio Uteri, n (%)	8 (9.8%)	10 (12%)	1.22 (0.85–1.39)	0.72
Surgical Site Infection, n (%)	2 (12.5%)	2 (13.3%)	1.06 (0.75–1.28)	0.22

Abbreviations: PGE—prostaglandin E; CRB—cervical ripening balloon; n—number; OR—odds ratio; CI—confidence interval; SD—standard deviation; CS—cesarean section.

**Table 3 jcm-11-02138-t003:** Neonatal outcomes.

	PGE2 (n = 81)	CRB (n = 83)	OR (95% CI)	*p* Value
5-min Apgar < 5, n (%)	3 (3.7)	3 (3.6)	1.02 (0.89–1.23)	0.78
Arterial cord pH < 7.1, n (%)	7 (8.6)	9 (10.8)	1.25 (0.82–1.46)	0.64
Admission to NICU, n (%)	2 (2.4)	2 (2.4)	1.13 (0.75–1.38)	0.77
Neonatal sepsis, n (%)	1 (1.2)	1 (1.2)	1.19 (0.68–1.84)	0.71
Need for oxygen supplementation, n (%)	4 (4.9)	5 (6)	1.22 (0.84–1.51)	0.69
Erb’s palsy, n (%)	1 (1.2)	1 (1.2)	1.01 (0.87–1.19)	0.86
Neonatal hypoglycemia, n (%)	3 (3.7)	3 (3.6)	1.02 (0.93–1.12)	0.92

Abbreviations: PGE—prostaglandin E; CRB—cervical ripening balloon; n—number; OR—odds ratio; CI—confidence interval; NICU—neonatal intensive care unit.

## Data Availability

The data supporting our conclusion can be obtained from r_lauterbach@rambam.health.gov.il.

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
