# Peer review of "Vaginal Dinoprostone Insert versus Cervical Ripening Balloon for Term Induction of Labor in Obese Nulliparas—A Randomized Controlled Trial"

_jcm, 2022, doi:10.3390/jcm11082138_

Round 1

Reviewer 1 Report

The authors present the findings of a randomized controlled trial of vaginal prostaglandin compared to cervical ripening balloon (CRB) for term induction of 164 obese nulliparas. CRB induction was associated with a reduced delivery rate <24 hours, shorter time to deliver, decreased rate of tachysystole and higher maternal satisfaction and lower maternal anxiety. There was no difference in CS rate or maternal or perinatal outcomes. The authors conclude that further research is warranted to establish the preferred IOL method for nulliparous obese women.

Comments:

1) This study is a well conducted and clearly described single-centre RCT comparing the efficacy of vaginal prostaglandin and cervical ripening balloon for term induction of obese nulliparas. Given the increased morbidity, higher rate of failed induction and increased CS rate of obese pregnant women, the findings of this study will be of interest to obstetric clinicians and academics. 

Abstract:

2) I would suggest including the significantly reduced tachysystole rate associated with CRB use with the study results. I would also suggest replacing "no difference in vaginal delivery rates" with "no difference in CS rates" since CS rates associated with method of IOL in this population is what the reader is primarily interested in.

Materials and Methods:

3) The study was conducted over a four year time frame to achieve the estimated sample size. Were there changes in clinical practice or provider bias in IOL method that may have affected recruitment and results?

4) Randomization utilized a computer generated allocation with a 1:1 ratio. What measures (?block size) were taken to ensure that providers and participants were not able to know their assignment ahead of time and how might that knowledge have affected recruitment? 

5) The primary outcome was delivery with 24 hours with a 20% reduction in CRB-treated participants used to calculate the sample size. Is this difference in treatment effect clinically relevant?

6) Although references are provided, confirmation of validity and internal reliability of the participant satisfaction and anxiety measures in the text would be helpful.

Results:

7) Table 1: In randomized controlled trials, p-values are generally not required for participant characteristics since randomization should effectively eliminate significant differences in demographic and other baseline characteristics.

8) Though not specially stated in the results section or Table 2, there is no difference in CS rates between groups (PGE2 16(19.8%), CRB 15(18.1%). I would suggest that this outcome be included, either in addition to vaginal delivery rates or as a replacement. The CS rate associated with IOL methods is the primary focus of practitioners and should be included as such.

9) I would suggest that the sample size is too small to draw attention to differences between groups regarding indications for CS. These differences could simpley be the result of multiple comparisons.

Discussion:

10) The increased satisfaction and decreased anxiety is presented but not explained. Some speculation regarding the reasons for these findings among obese nulliparas undergoing CRB induction would be useful.

11) Though not specific to this study population, the relatively neutral findings of the Cochrane systematic review of RCTs comparing vaginal PGE2 and balloon (1685 - 6619 women depending on primary outcome) should be noted in the Discussion and referenced.

12) Though blinding, as pointed out by the authors, was difficult to impossible, the potential effect of knowledge of method of IOL on provider intervention bias and maternal satisfaction and anxiety response to vaginal PGE2 vs CRB should be addressed.

Author Response

Dear Editor,

Attached please find our revised paper entitled “Vaginal dinoprostone insert versus cervical ripening balloon for term induction of labor in obese nulliparas-A randomized controlled trial”.

We hope the reviewers and editorial team are satisfied by our response to their knowledgeful comments. We believe their comments have upgraded the level of our manuscript and thank them for their time and effort.

Our response to the reviewers appears on the following pages.

Thank you in advance for considering our revised paper for publication in the Journal of Clinical Medicine.

  1. This study is a well conducted and clearly described single-centre RCT comparing the efficacy of vaginal prostaglandin and cervical ripening balloon for term induction of obese nulliparas. Given the increased morbidity, higher rate of failed induction and increased CS rate of obese pregnant women, the findings of this study will be of interest to obstetric clinicians and academics. 

Thank you.

  1. I would suggest including the significantly reduced tachysystole rate associated with CRB use with the study results. I would also suggest replacing "no difference in vaginal delivery rates" with "no difference in CS rates" since CS rates associated with method of IOL in this population is what the reader is primarily interested in.

The abstract was amended as suggested.

  1. The study was conducted over a four year time frame to achieve the estimated sample size. Were there changes in clinical practice or provider bias in IOL method that may have affected recruitment and results?

There were no changes in practice over the study period that may have caused bias for any of the 2 IOL methods.

  1. Randomization utilized a computer generated allocation with a 1:1 ratio. What measures (?block size) were taken to ensure that providers and participants were not able to know their assignment ahead of time and how might that knowledge have affected recruitment?
    Block size was added to methods section as suggested.
  2. The primary outcome was delivery with 24 hours with a 20% reduction in CRB-treated participants used to calculate the sample size. Is this difference in treatment effect clinically relevant?

The authors believe that due to the high volume of inductions (2-3 times higher) since the publication of the ARRIVE trial and sequential studies, the time to delivery has become a grave issue in evaluating treatment efficacy. Furthermore, shorter intervals to delivery have been shown to be associated with shorter intervals to delivery. This is why this parameter is extremely clinically relevant in any current IOL study.

  1. Although references are provided, confirmation of validity and internal reliability of the participant satisfaction and anxiety measures in the text would be helpful.

A figure has been added to further validate the results of the patient satisfaction and anxiety questionnaire scores.

  1. Table 1: In randomized controlled trials, p-values are generally not required for participant characteristics since randomization should effectively eliminate significant differences in demographic and other baseline characteristics.

P-values were omitted as suggested.

  1. Though not specially stated in the results section or Table 2, there is no difference in CS rates between groups (PGE2 16(19.8%), CRB 15(18.1%). I would suggest that this outcome be included, either in addition to vaginal delivery rates or as a replacement. The CS rate associated with IOL methods is the primary focus of practitioners and should be included as such.

Total CS rates were added to table 2 as suggested.

  1. I would suggest that the sample size is too small to draw attention to differences between groups regarding indications for CS. These differences could simply be the result of multiple comparisons.

While we agree with the reviewer on some level, the differences in CS rate by indication

  1. The increased satisfaction and decreased anxiety is presented but not explained. Some speculation regarding the reasons for these findings among obese nulliparas undergoing CRB induction would be useful.

Investigator`s speculation has been added to the discussion as suggested.

  1. Though not specific to this study population, the relatively neutral findings of the Cochrane systematic review of RCTs comparing vaginal PGE2 and balloon (1685 - 6619 women depending on primary outcome) should be noted in the Discussion and referenced.

The reference and findings were added to the proper sections of the manuscript as suggested.

  1. Though blinding, as pointed out by the authors, was difficult to impossible, the potential effect of knowledge of method of IOL on provider intervention bias and maternal satisfaction and anxiety response to vaginal PGE2 vs CRB should be addressed.

We addressed the possible effect of non-blinding on validity of satisfaction and anxiety as suggested.

Reviewer 2 Report

In this  study, Lauterbach et al compared mechanical induction of labor to pharmacological induction of labor in nulliparas obese women as a controlled randomized study. 

Current results reveal that IOL with CRB compared to PGE2 is basically associated with reduced  delivery time and  higher patient satisfaction. Please see my comments/concerns below;

  1. Although results of this study provide important data, this reviewer has minor concerns about the design of the groups. This study is designed on an obese pregnant population. Since obesity is strongly linked to metabolic syndrome, these patients included in this study may have very different metabolic profiles such as altered lipid profile, glycemic profile, or altered insulin metabolism which may eventually effect the response to the mentioned IOL techniques.  A better separation of groups including metabolic profile may give more accurate results.
  2.  Authors mentioned in the study design that, removal of the CRB was performed 12 hours after insertion and  removal of the dinoprostone vaginal insert was performed 24 hours after insertion if self-expulsion did not occur.  What was the ratio of self-expulsion in this study and what would be the authors comments if there is different time points of self-expulsions in both techniques? 
  3. Authors limited the discussion section with the previous data related  obese pregnancies only. Including the non-obese data related with same/similar IOL techniques will provide a stronger discussion and a better understanding of current data. 

Author Response

Dear Editor,

Attached please find our revised paper entitled “Vaginal dinoprostone insert versus cervical ripening balloon for term induction of labor in obese nulliparas-A randomized controlled trial”.

We hope the reviewers and editorial team are satisfied by our response to their knowledgeful comments. We believe their comments have upgraded the level of our manuscript and thank them for their time and effort.

Our response to the reviewers appears on the following pages.

Thank you in advance for considering our revised paper for publication in the Journal of Clinical Medicine.

  1. Although results of this study provide important data, this reviewer has minor concerns about the design of the groups. This study is designed on an obese pregnant population. Since obesity is strongly linked to metabolic syndrome, these patients included in this study may have very different metabolic profiles such as altered lipid profile, glycemic profile, or altered insulin metabolism which may eventually effect the response to the mentioned IOL techniques. A better separation of groups including metabolic profile may give more accurate results.

While we agree with the reviewer`s remarks a better separation of the groups may not be obtained and may be clinically inaccurate in pregnancy as lipid profiles are affected significantly by pregnancy and are not routinely tested in pregnancy as international guidelines underline them obsolete. Regarding glycemic profiles, since the rates of diabetes in pregnancy were not significantly different between the 2 induction groups, the glycemic profile of the remainder of the study population that underwent routine diabetes screening in pregnancy has not been shown in previous studies to affect induction success. Furthermore, the majority of previous studies regrading IOL in obese populations have not found significant differences in non-diabetic patients with obesity in terms of IOL efficacy. We hope our reply addresses the reviewer`s concerns.

  1. Authors mentioned in the study design that, removal of the CRB was performed 12 hours after insertion and  removal of the dinoprostone vaginal insert was performed 24 hours after insertion if self-expulsion did not occur.  What was the ratio of self-expulsion in this study and what would be the authors comments if there is different time points of self-expulsions in both techniques? 

The authors didn`t address the rate of self expulsion due to the fact that when this occurs after insertion of a CRB compared to a PGE2 insert there are different meanings to the premature expulsion. While a self expulsion of a CRB is considered a sign that the patient`s cervix is dilated enough for the CRB to be expelled, and the patient may be in active labor, in fact, thus pointing at a successful induction, self expulsion of a vaginal PGE2 insert that is merely located in the vagina, not being held by the tension of any tissue (unlike the CRB that is inflated inside the cervix and cannot be remove without deflating or dilatation), is a technical matter, sometimes occurring by the patient by mistake and sometimes due to ill-insertion. In any matter, if the CRB is expelled the CRB is not replaced by an additional CRB but if the PGE2 vaginal insert is expelled it is inserted once again by a physician. For these reasons, the self-expulsion rates of PGE2 are irrelevant to any time points of self expulsion, and the expulsion rates of CRBs by time points are again less relevant. We hope our reply addresses the reviewer`s concerns.

  1. Authors limited the discussion section with the previous data related  obese pregnancies only. Including the non-obese data related with same/similar IOL techniques will provide a stronger discussion and a better understanding of current data. 

The proper changes have been made as suggested and results of a Cochrane review were added in the 2nd paragraph of the discussion with a relevant reference.